# Compatible Reward Inverse Reinforcement Learning

**Alberto Maria Metelli**
DEIB
Politecnico di Milano, Italy
`albertomaria.metelli@polimi.it`

**Matteo Pirotta**
SequeL Team
Inria Lille, France
`matteo.pirotta@inria.fr`

**Marcello Restelli**
DEIB
Politecnico di Milano, Italy
`marcello.restelli@polimi.it`

## Abstract

Inverse Reinforcement Learning (IRL) is an effective approach to recover a reward function that explains the behavior of an expert by observing a set of demonstrations. This paper is about a novel model-free IRL approach that, differently from most of the existing IRL algorithms, does not require to specify a function space where to search for the expert's reward function. Leveraging on the fact that the policy gradient needs to be zero for any optimal policy, the algorithm generates a set of basis functions that span the subspace of reward functions that make the policy gradient vanish. Within this subspace, using a second-order criterion, we search for the reward function that penalizes the most a deviation from the expert's policy. After introducing our approach for finite domains, we extend it to continuous ones. The proposed approach is empirically compared to other IRL methods both in the (finite) Taxi domain and in the (continuous) Linear Quadratic Gaussian (LQG) and Car on the Hill environments.

## 1   Introduction

Imitation learning aims to learn to perform a task by observing only expert's demonstrations. We consider the settings where only expert's demonstrations are given, no information about the dynamics and the objective of the problem is provided (e.g., reward) or ability to query for additional samples. The main approaches solving this problem are behavioral cloning [1] and inverse reinforcement learning [2]. The former recovers the demonstrated policy by learning the state-action mapping in a supervised learning way, while inverse reinforcement learning aims to learn the reward function that makes the expert optimal. Behavioral Cloning (BC) is simple, but its main limitation is the intrinsic goal, i.e., to replicate the observed policy. This task has several limitations: it requires a huge amount of data when the environment (or the expert) is stochastic [3]; it does not provide good generalization or a description of the expert's goal. On the contrary, Inverse Reinforcement Learning (IRL) accounts for generalization and transferability by directly learning the reward function. This information can be transferred to any new environment in which the features are well defined. As a consequence, IRL allows recovering the optimal policy a posteriori, even under variations of the environment. IRL has received a lot of attention in literature and has succeeded in several applications [e.g., 4, 5, 6, 7, 8]. However, BC and IRL are tightly related by the intrinsic relationship between reward and optimal policy. The reward function defines the space of optimal policies and to recover the reward it is required to observe/recover the optimal policy. The idea of this paper, and of some recent paper [e.g., 9, 8, 3], is to exploit the synergy between BC and IRL.

Unfortunately, also IRL approaches present issues. First, several IRL methods require solving the forward problem as part of an inner loop [e.g., 4, 5]. Literature has extensively focused on removing this limitation [10, 11, 9] in order to scale IRL to real-world applications [12, 3, 13]. Second, IRL methods generally require designing the function space by providing *features* that capture the structure of the reward function [e.g., 4, 14, 5, 10, 15, 9]. This information, provided in addition to expert's demonstrations, is critical for the success of the IRL approach. The issue of designing the function

space is a well-known problem in supervised learning, but it is even more critical in IRL since a wrong choice might prevent the algorithm from finding good solutions to the IRL problem [2, 16], especially when linear reward models are considered. The importance of incorporating feature construction in IRL has been known in literature since a while [4] but, as far as we know, it has been explicitly addressed only in [17]. Recently, IRL literature, by mimicking supervised learning one, has focused on exploiting neural network capability of automatically constructing relevant features out of the provided data [12, 8, 13]. By exploiting a "black-box" approach, these methods do not take advantage of the structure of the underlying Markov decision process (in the phase of feature construction).

We present an IRL algorithm that constructs reward features directly from expert's demonstrations. The proposed algorithm is model-free and does not require solving the forward problem (i.e., finding an optimal policy given a candidate reward function) as an inner step. The Compatible Reward Inverse Reinforcement Learning (CR-IRL) algorithm builds a reward function that is *compatible* with the expert's policy. It mixes BC and IRL in order to recover the "optimal" and most "informative" reward function in the space spanned by the recovered features. Inspired by the gradient-minimization IRL approach proposed in [9], we focus on the space of reward functions that makes the policy gradient of the expert vanish. Since a zero gradient is only a necessary condition for optimality, we consider a second order optimality criterion based on the policy Hessian to rank the reward functions and finally select the best one (i.e., the one that penalizes the most a deviation from the expert's policy).

## 2 Algorithm Overview

A Markov Decision Process (MDP) [18] is defined as $\mathcal{M} = (\mathcal{S}, \mathcal{A}, \mathcal{P}, R, \gamma, \mu)$ where $\mathcal{S}$ is the state space, $\mathcal{A}$ is the action space, $\mathcal{P}(s'|s, a)$ is a Markovian transition model that defines the conditional distribution of the next state $s'$ given the current state $s$ and the current action $a$, $\gamma \in [0, 1]$ is the discount factor, $R(s, a)$ is the expected reward for performing action $a$ in state $s$ and $\mu$ is the distribution of the initial state. The optimal policy $\pi^*$ is the policy that maximizes the discounted sum of rewards $\mathbb{E}[\sum_{t=0}^{+\infty} \gamma^t R(s_t, a_t)|\pi, \mathcal{M}]$.

CR-IRL takes as input a parametric policy space $\Pi_\Theta = \{\pi_{\boldsymbol{\theta}} : \boldsymbol{\theta} \in \Theta \subseteq \mathbb{R}^k\}$ and a set of rewardless trajectories from the expert policy $\pi^E$, denoted by $\mathcal{D} = \{(s_{\tau_i,0}, a_{\tau_i,0}, \ldots, s_{\tau_i,T(\tau_i)}, a_{\tau_i,T(\tau_i)})\}$, where $s_{\tau_i,t}$ is the $t$-th state in trajectory $\tau_i$ and $i = 1, \ldots, N$. CR-IRL is a non-iterative algorithm that recovers a reward function for which the expert is optimal without requiring to specify a reward function space. It starts building the features $\{\phi_i\}$ of the value function that are compatible with policy $\pi^E$, i.e., that make the policy gradient vanish (Phase 1, see Sec. 3). This step requires a parametric representation $\pi_{\boldsymbol{\theta}^E} \in \Pi_\Theta$ of the expert's policy which can be obtained through behavioral cloning.[1] The choice of the policy space $\Pi_\Theta$ influences the size of the functional space used by CR-IRL for representing the value function (and the reward function) associated with the expert's policy. In order to formalize this notion, we introduce the *policy rank*, a quantity that represents the ability of a parametric policy to reduce the dimensions of the approximation space for the value function of the expert's policy. Once these value features have been built, they can be transformed into reward features $\{\psi_i\}$ (Phase 2 see Sec. 4) by means of the Bellman equation [18] (model-based) or reward shaping [19] (model-free). All the rewards spanned by the features $\{\psi_i\}$ satisfy the first-order necessary optimality condition [20], but we are not sure about their nature (minima, maxima or saddle points). The final step is thus to recover a reward function that is maximized by the expert's policy (Phase 3 see Sec. 5). This is achieved by considering a second-order optimality condition, with the idea that we want the reward function that penalizes the most a deviation from the parameters of the expert's policy $\pi_{\boldsymbol{\theta}^E}$. This criterion is similar in spirit to what done in [2, 4, 14], where the goal is to identify the reward function that makes the expert's policy better than any other policy by a margin. The algorithmic structure is reported in Alg. 1.

IRL literature usually considers two different settings: optimal or sub-optimal expert. This distinction is necessary when a fixed reward space is provided. In fact, the demonstrated behavior may not be optimal under the considered reward space. In this case, the problem becomes somehow not well defined and additional "optimality" criteria are required [16]. This is not the case for CR-IRL that is able to automatically generate the space of reward functions that make the policy gradient vanish,

thus containing also reward functions under which the recovered expert's policy $\pi_{\boldsymbol{\theta}^E}$ is optimal. In the rest of the paper, we will assume to have a parametric representation of the expert's policy that we will denote for simplicity by $\pi_{\boldsymbol{\theta}}$.

# 3   Expert's Compatible Value Features

In this section, we present the procedure to obtain the set $\{\phi_i\}_{i=1}^p$ of Expert's COmpatible Q-features (ECO-Q) that make the policy gradient vanish[2] (Phase 1). We start introducing the policy gradient and the associated first-order optimality condition. We will indicate with $\mathbb{T}$ the set of all possible trajectories, $p_{\boldsymbol{\theta}}(\tau)$ the probability density of trajectory $\tau$ and $R(\tau)$ the $\gamma$-discounted trajectory reward defined as $R(\tau) = \sum_{t=0}^{T(\tau)} \gamma^t R(s_{\tau,t}, a_{\tau,t})$ that, in our settings, is obtained as a linear combination of reward features. Given a policy $\pi_{\boldsymbol{\theta}}$, the expected $\gamma$-discounted return for an infinite horizon MDP is:

$$J(\boldsymbol{\theta}) = \int_{\mathcal{S}} d_{\mu}^{\pi_{\boldsymbol{\theta}}}(s) \int_{\mathcal{A}} \pi_{\boldsymbol{\theta}}(a|s) R(s,a) \mathrm{d}a \mathrm{d}s = \int_{\mathbb{T}} p_{\boldsymbol{\theta}}(\tau) R(\tau) \mathrm{d}\tau,$$

where $d_{\mu}^{\pi_{\boldsymbol{\theta}}}$ is the $\gamma$-discounted future state occupancy [21]. If $\pi_{\boldsymbol{\theta}}$ is differentiable w.r.t. the parameter $\boldsymbol{\theta}$, the gradient of the expected reward (*policy gradient*) [21, 22, 23] is:

$$\nabla_{\boldsymbol{\theta}} J(\boldsymbol{\theta}) = \int_{\mathcal{S}} \int_{\mathcal{A}} d_{\mu}^{\pi_{\boldsymbol{\theta}}}(s,a) \nabla_{\boldsymbol{\theta}} \log \pi_{\boldsymbol{\theta}}(a|s) Q^{\pi_{\boldsymbol{\theta}}}(s,a) \mathrm{d}a \mathrm{d}s = \int_{\mathbb{T}} p_{\boldsymbol{\theta}}(\tau) \nabla_{\boldsymbol{\theta}} \log p_{\boldsymbol{\theta}}(\tau) R(\tau) \mathrm{d}\tau, \quad (1)$$

where $d_{\mu}^{\pi_{\boldsymbol{\theta}}}(s,a) = d_{\mu}^{\pi_{\boldsymbol{\theta}}}(s)\pi_{\boldsymbol{\theta}}(a|s)$ is the $\gamma$-discounted future state-action occupancy, which represents the expected discounted number of times action $a$ is executed in state $s$ given $\mu$ as initial state distribution and following policy $\pi_{\boldsymbol{\theta}}$. When $\pi_{\boldsymbol{\theta}}$ is an optimal policy in the class of policies $\Pi_{\Theta} = \{\pi_{\boldsymbol{\theta}} : \boldsymbol{\theta} \in \Theta \subseteq \mathbb{R}^k\}$ then $\boldsymbol{\theta}$ is a *stationary point* of the expected return and thus $\nabla_{\boldsymbol{\theta}} J(\boldsymbol{\theta}) = \mathbf{0}$ (*first-order necessary conditions for optimality* [20]).

We assume the space $\mathcal{S} \times \mathcal{A}$ to be a Hilbert space [24] equipped with the weighted inner product:[3]

$$\langle f, g \rangle_{\mu, \pi_{\boldsymbol{\theta}}} = \int_{\mathcal{S}} \int_{\mathcal{A}} f(s,a) d_{\mu}^{\pi_{\boldsymbol{\theta}}}(s,a) g(s,a) \mathrm{d}s \mathrm{d}a. \quad (2)$$

When $\pi_{\boldsymbol{\theta}}$ is optimal for the MDP, $\nabla_{\boldsymbol{\theta}} \log \pi_{\boldsymbol{\theta}}$ and $Q^{\pi_{\boldsymbol{\theta}}}$ are *orthogonal* w.r.t. the inner product (2). We can exploit the orthogonality property to build an approximation space for the Q-function. Let $G_{\pi_{\boldsymbol{\theta}}} = \{\nabla_{\boldsymbol{\theta}} \log \pi_{\boldsymbol{\theta}} \boldsymbol{\alpha} : \boldsymbol{\alpha} \in \mathbb{R}^k\}$ the subspace spanned by the gradient of the log-policy $\pi_{\boldsymbol{\theta}}$. From equation (1) finding an approximation space for the Q-function is equivalent to find the orthogonal complement of the subspace $G_{\pi_{\boldsymbol{\theta}}}$, which in turn corresponds to find the null space of the functional:

$$\mathcal{G}_{\pi_{\boldsymbol{\theta}}}[\phi] = \langle \nabla_{\boldsymbol{\theta}} \log \pi_{\boldsymbol{\theta}}, \phi \rangle_{\mu, \pi_{\boldsymbol{\theta}}}. \quad (3)$$

We define an *Expert's COmpatible Q-feature* as any function $\phi$ making the functional (3) null. This space $G_{\pi_{\boldsymbol{\theta}}}^{\perp} := \mathrm{null}(\mathcal{G}_{\pi_{\boldsymbol{\theta}}})$ represents the Hilbert subspace of the features for the Q-function that are compatible with the policy $\pi_{\boldsymbol{\theta}}$ in the sense that any Q-function optimized by policy $\pi_{\boldsymbol{\theta}}$ can be expressed as a linear combination of those features. Section 3.2 and 3.3 describe how to compute the ECO-Q from samples in finite and continuous MDPs, respectively. The dimension of $G_{\pi_{\boldsymbol{\theta}}}^{\perp}$ is typically very large since the number $k$ of policy parameters is significantly smaller than the number of state-action pairs. A formal discussion of this issue for finite MDPs is presented in the next section.

## 3.1   Policy rank

The parametrization of the expert's policy influences the size of $G_{\pi_{\boldsymbol{\theta}}}^{\perp}$. Intuition suggests that the larger the number $k$ of the parameters the more the policy is *informative* to infer the Q-function and so the reward function. This is motivated by the following rationale. Consider representing the expert's policy using two different policy models such that one model is a superclass of the other one (for instance, assume to use linear models where the features used in the simpler model are a subset of the features used by policies in the other model). All the reward functions that make the policy gradient

vanish with the rich policy model, do the same with the simpler model, while the vice versa does not hold. This suggests that complex policy models are able to reduce more the space of optimal reward function w.r.t. simpler models. This notion plays an important role for finite MDPs, i.e., MDPs where the state-action space is finite. We formalize the ability of a policy to infer the characteristics of the MDP with the concept of *policy rank*.

**Definition 1.** *Let $\pi_{\boldsymbol{\theta}}$ a policy with $k$ parameters belonging to the class $\Pi_{\Theta}$ and differentiable in $\boldsymbol{\theta}$. The policy rank is the dimension of the space of the linear combinations of the partial derivatives of $\pi_{\boldsymbol{\theta}}$ w.r.t. $\boldsymbol{\theta}$:*

$$\mathrm{rank}(\pi_{\boldsymbol{\theta}}) = \dim(\Gamma_{\pi_{\boldsymbol{\theta}}}), \quad \Gamma_{\pi_{\boldsymbol{\theta}}} = \{\nabla_{\boldsymbol{\theta}}\pi_{\boldsymbol{\theta}}\boldsymbol{\alpha} : \boldsymbol{\alpha} \in \mathbb{R}^k\}.$$

A first important note is that the policy rank depends not only on the policy model $\Pi_{\Theta}$ but also on the *value* of the parameters of the policy $\pi_{\boldsymbol{\theta}}$. So the policy rank is a property of the *policy* not of the *policy model*. The following bound on the policy rank holds (the proof can be found in App. A.1).

**Proposition 1.** *Given a finite MDP $\mathcal{M}$, let $\pi_{\boldsymbol{\theta}}$ a policy with $k$ parameters belonging to the class $\Pi_{\Theta}$ and differentiable in $\boldsymbol{\theta}$, then:* $\mathrm{rank}(\pi_{\boldsymbol{\theta}}) \leq \min\{k, |\mathcal{S}||\mathcal{A}| - |\mathcal{S}|\}$.

From an intuitive point of view this is justified by the fact that $\pi_{\boldsymbol{\theta}}(\cdot|s)$ is a probability distribution. As a consequence, for all $s \in \mathcal{S}$ the probabilities $\pi_{\boldsymbol{\theta}}(a|s)$ must sum up to 1, removing $|\mathcal{S}|$ degrees of freedom. This has a relevant impact on the algorithm since it induces a lower bound on the dimension of the orthogonal complement $\dim(G_{\pi_{\boldsymbol{\theta}}}^{\perp}) \geq \max\{|\mathcal{S}||\mathcal{A}| - k, |\mathcal{S}|\}$, thus even the most flexible policy (i.e., a policy model with a parameter for each state-action pair) cannot determine a unique reward function that makes the expert's policy optimal, leaving $|\mathcal{S}|$ degrees of freedom. It follows that it makes no sense to consider a policy with more than $|\mathcal{S}||\mathcal{A}| - |\mathcal{S}|$ parameters. The generalization capabilities enjoyed by the recovered reward function are deeply related to the choice of the policy model. Complex policies (many parameters) would require finding a reward function that explains the value of all the parameters, resulting in a possible overfitting, whereas a simple policy model (few parameters) would enforce generalization as the imposed constraints are fewer.

## 3.2 Construction of ECO-Q in Finite MDPs

We now develop in details the algorithm to generate ECO-Q in the case of finite MDPs. From now on we will indicate with $|\mathcal{D}|$ the number of distinct state-action pairs *visited* by the expert along the available trajectories. When the state-action space is finite the inner product (2) can be written in matrix notation as:

$$\langle \mathbf{f}, \mathbf{g} \rangle_{\mu, \pi_{\boldsymbol{\theta}}} = \mathbf{f}^T \mathbf{D}_{\mu}^{\pi_{\boldsymbol{\theta}}} \mathbf{g},$$

where $\mathbf{f}$, $\mathbf{g}$ and $\mathbf{d}_{\mu}^{\pi_{\boldsymbol{\theta}}}$ are real vectors with $|\mathcal{D}|$ components and $\mathbf{D}_{\mu}^{\pi_{\boldsymbol{\theta}}} = \mathrm{diag}(\mathbf{d}_{\mu}^{\pi_{\boldsymbol{\theta}}})$. The term $\nabla_{\boldsymbol{\theta}} \log \pi_{\boldsymbol{\theta}}$ is a $|\mathcal{D}| \times k$ real matrix, thus finding the null space of the functional (3) is equivalent to finding the null space of the matrix $\nabla_{\boldsymbol{\theta}} \log \pi_{\boldsymbol{\theta}}^T \mathbf{D}_{\mu}^{\pi_{\boldsymbol{\theta}}}$. This can be done for instance through SVD which allows to obtain a set of orthogonal basis functions $\boldsymbol{\Phi}$. Given that the weight vector $d_{\mu}^{\pi_{\boldsymbol{\theta}}}(s, a)$ is usually unknown, it needs to be estimated. Since the policy $\pi_{\boldsymbol{\theta}}$ is known, we need to estimate just $d_{\mu}^{\pi_{\boldsymbol{\theta}}}(s)$, as $d_{\mu}^{\pi_{\boldsymbol{\theta}}}(s, a) = d_{\mu}^{\pi_{\boldsymbol{\theta}}}(s)\pi_{\boldsymbol{\theta}}(a|s)$. A Monte Carlo estimate exploiting the expert's demonstrations in $\mathcal{D}$ is:

$$\hat{d}_{\mu}^{\pi_{\boldsymbol{\theta}}}(s) = \frac{1}{N} \sum_{i=1}^{N} \sum_{t=0}^{T(\tau_i)} \gamma^t \mathbb{1}(s_{\tau_i, t} = s). \tag{4}$$

## 3.3 Construction of ECO-Q in Continuous MDPs

To extend the previous approach to the continuous domain we assume that the state-action space is equipped with the Euclidean distance. Now we can adopt an approach similar to the one exploited to extend Proto-Value Functions (PVF) [25, 26] to infinite observation spaces [27]. The problem is treated as a discrete one considering only the state-action pairs visited along the collected trajectories. A Nyström interpolation method is used to approximate the value of a feature in a non-visited state-action pair as a weighted mean of the values of the closest $k$ features. The weight of each feature is computed by means of a Gaussian kernel placed over the Euclidean space $\mathcal{S} \times \mathcal{A}$:

$$\mathcal{K}\big((s, a), (s', a')\big) = \exp\Big(-\frac{1}{2\sigma_{\mathcal{S}}^2}\|s - s'\|_2^2 - \frac{1}{2\sigma_{\mathcal{A}}^2}\|a - a'\|_2^2\Big), \tag{5}$$

where $\sigma_{\mathcal{S}}$ and $\sigma_{\mathcal{A}}$ are respectively the state and action bandwidth. In our setting this approach is fully equivalent to a kernel k-Nearest Neighbors regression.

# 4 Expert's Compatible Reward Features

The set of ECO-Q basis functions allows representing the optimal value function under the policy $\pi_{\boldsymbol{\theta}}$. In this section, we will show how it is possible to exploit ECO-Q functions to generate basis functions for the reward representation (Phase 2). In principle, we can use the Bellman equation to obtain the reward from the Q-function but this approach requires the knowledge of the transition model (see App. B). The reward can be recovered in a model-free way by exploiting optimality-invariant reward transformations.

Reversing the Bellman equation [e.g., 10] allows finding the reward space that generates the estimated Q-function. However, IRL is interested in finding just a reward space under which the expert's policy is optimal. This problem can be seen as an instance of reward shaping [19] where the authors show that the space of all the reward functions sharing the same optimal policy is given by:

$$R'(s,a) = R(s,a) + \gamma \int_{\mathcal{S}} \mathcal{P}(s'|s,a)\chi(s')\mathrm{d}s' - \chi(s),$$

where $\chi(s)$ is a state-dependent potential function. A smart choice [19] is to set $\chi = V^{\pi_{\boldsymbol{\theta}}}$ under which the new reward space is given by the advantage function: $R'(s,a) = Q^{\pi_{\boldsymbol{\theta}}}(s,a) - V^{\pi_{\boldsymbol{\theta}}}(s) = A^{\pi_{\boldsymbol{\theta}}}(s,a)$. Thus the expert's advantage function is an admissible reward optimized by the expert's policy itself. This choice is, of course, related to using $Q^{\pi_{\boldsymbol{\theta}}}$ as reward. However, the advantage function encodes a more local and more transferable information w.r.t. the Q-function.

The space of reward features can be recovered through matrix equality $\boldsymbol{\Psi} = (\mathbf{I} - \tilde{\boldsymbol{\pi}}_{\boldsymbol{\theta}})\boldsymbol{\Phi}$, where $\tilde{\boldsymbol{\pi}}_{\boldsymbol{\theta}}$ is a $|\mathcal{D}| \times |\mathcal{D}|$ matrix obtained from $\boldsymbol{\pi}_{\boldsymbol{\theta}}$ repeating the row of each visited state a number of times equal to the number of distinct actions performed by the expert in that state. Notice that this is a simple linear transformation through the expert's policy. The specific choice of the state-potential function has the advantage to improve the learning capabilities of any RL algorithm [19]. This is not the only choice of the potential function possible, but it has the advantage of allowing model-free estimation.

Once the ECO-R basis functions have been generated, they can be used to feed any IRL algorithm that represents the expert's reward through a linear combination of basis functions. In the next section, we propose a new method based on the optimization of a second-order criterion that favors reward functions that significantly penalize deviations from the expert's policy.

# 5 Reward Selection via Second-Order Criterion

Any linear combination of the ECO-R $\{\psi_i\}_{i=1}^p$ makes the gradient vanish, however in general this is not sufficient to ensure that the policy parameter $\boldsymbol{\theta}$ is a maximum of $J(\boldsymbol{\theta})$. Combinations that lead to minima or saddle points should be discarded. Furthermore, provided that a subset of ECO-R leading to maxima has been selected, we should identify a single reward function in the space spanned by this subset of features (Phase 3). Both these requirements can be enforced by imposing a second-order optimality criterion based on the policy Hessian that is given by [28, 29]:

$$\mathcal{H}_{\boldsymbol{\theta}} J(\boldsymbol{\theta}, \boldsymbol{\omega}) = \int_{\mathbb{T}} p_{\boldsymbol{\theta}}(\tau) \Big( \nabla_{\boldsymbol{\theta}} \log p_{\boldsymbol{\theta}}(\tau) \nabla_{\boldsymbol{\theta}} \log p_{\boldsymbol{\theta}}(\tau)^T + \mathcal{H}_{\boldsymbol{\theta}} \log p_{\boldsymbol{\theta}}(\tau) \Big) R(\tau, \boldsymbol{\omega}) \mathrm{d}\tau,$$

where $\boldsymbol{\omega}$ is the reward weight and $R(\tau, \boldsymbol{\omega}) = \sum_{i=1}^p \omega_i \sum_{t=0}^{T(\tau)} \gamma^t \psi_i(s_{\tau,t}, a_{\tau,t})$.

In order to retain only maxima we need to impose that the Hessian is negative definite. Furthermore, we aim to find the reward function that best represents the optimal policy parametrization in the sense that even a slight change of the parameters of the expert's policy induces a significant degradation of the performance. Geometrically this corresponds to find the reward function for which the expected return locally represents the sharpest hyper-paraboloid. These requirements can be enforced using a Semi-Definite Programming (SDP) approach where the objective is to *minimize the maximum eigenvalue* of the Hessian whose eigenvector corresponds to the direction of minimum curvature (*maximum eigenvalue optimality* criterion). This problem is not appealing in practice due to its high computational burden. Furthermore, it might be the case that the strict negative definiteness constraint is never satisfied due to blocked-to-zero eigenvalues (for instance in presence of policy parameters that do not affect the policy performance). In these cases, we can consider maximizing an index of the overall concavity. The trace of the Hessian, being the sum of the eigenvalues, can be used for this purpose. This problem can be still defined as a SDP problem (*trace optimality* criterion). See App. C for details.

Trace optimality criterion, although less demanding w.r.t. the eigenvalue-based one, still displays performance degradation as the number of basis functions increases due to the negative definiteness constraint. Solving the semidefinite programming problem of one of the previous optimality criteria is unfeasible for almost all the real world problems. We are interested in formulating a non-SDP problem, which is a surrogate of the trace optimality criterion, that can be solved more efficiently (*trace heuristic* criterion). In our framework, the reward function can be expressed as a linear combination of the ECO-R so we can rewrite the Hessian as $\mathcal{H}_{\boldsymbol{\theta}} J(\boldsymbol{\theta}, \boldsymbol{\omega}) = \sum_{i=1}^{p} \omega_i \mathcal{H}_{\boldsymbol{\theta}} J_i(\boldsymbol{\theta})$ where $J_i(\boldsymbol{\theta})$ is the expected return considering as reward function $\psi_i$. We assume that the ECO-R are orthonormal in order to compare them.[4] The main challenge is how to select the weight $\boldsymbol{\omega}$ in order to get a (sub-)optimal trace minimizer that preserves the negative semidefinite constraint. From Weyl's inequality, we get a feasible solution by retaining only the ECO-Rs yield-

---

**Input:** $\mathcal{D} = \left\{ \left( s_{\tau_i,0}, a_{\tau_i,0}, \ldots, s_{\tau_i,T(\tau_i)}, a_{\tau_i,T(\tau_i)} \right) \right\}_{i=1}^{N}$ a set of expert's trajectories and parametric expert's policy $\pi_{\boldsymbol{\theta}}$.
**Output:** trace heuristic ECO-R, $R^{\text{tr}-\text{heu}}$.

**Phase 1**

1. Estimate $d_{\mu}^{\pi_{\boldsymbol{\theta}}}(s)$ for the visited state-action pairs using Eq. (4) and compute $d_{\mu}^{\pi_{\boldsymbol{\theta}}}(s,a) = d_{\mu}^{\pi_{\boldsymbol{\theta}}}(s)\pi_{\boldsymbol{\theta}}(a|s)$.
2. Collect $d_{\mu}^{\pi_{\boldsymbol{\theta}}}(s,a)$ in the $|\mathcal{D}| \times |\mathcal{D}|$ diagonal matrix $\mathbf{D}_{\mu}^{\pi_{\boldsymbol{\theta}}}$ and $\nabla_{\boldsymbol{\theta}} \log \pi_{\boldsymbol{\theta}}(s,a)$ in the $|\mathcal{D}| \times k$ matrix $\nabla_{\boldsymbol{\theta}} \log \pi_{\boldsymbol{\theta}}$.
3. Get the set of ECO-Q by computing the null space of matrix $\nabla_{\boldsymbol{\theta}} \log \pi_{\boldsymbol{\theta}}^T \mathbf{D}_{\mu}^{\pi_{\boldsymbol{\theta}}}$ through SVD: $\boldsymbol{\Phi} = \text{null}\left( \nabla_{\boldsymbol{\theta}} \log \pi_{\boldsymbol{\theta}}^T \mathbf{D}_{\mu}^{\pi_{\boldsymbol{\theta}}} \right)$.

4. Get the set of ECO-R by applying reward shaping to the set of ECO-Q: $\boldsymbol{\Psi} = (\mathbf{I} - \tilde{\boldsymbol{\pi}}_{\boldsymbol{\theta}})\boldsymbol{\Phi}$.
5. Apply SVD to orthogonalize $\boldsymbol{\Psi}$.

**Phase 2**

6. Estimate the policy Hessian for each ECO-R $\psi_i$, $i = 1, \ldots p$ using equation:[a]
$$\hat{\mathcal{H}}_{\boldsymbol{\theta}} J_i(\boldsymbol{\theta}) = \frac{1}{N} \sum_{j=1}^{N} \left( \nabla_{\boldsymbol{\theta}} \log p_{\boldsymbol{\theta}}(\tau_j) \nabla_{\boldsymbol{\theta}} \log p_{\boldsymbol{\theta}}(\tau_j)^T + \mathcal{H}_{\boldsymbol{\theta}} \log p_{\boldsymbol{\theta}}(\tau_j) \right) \left( \psi_i(\tau_j) - b \right).$$
7. Discard the ECO-R having indefinite Hessian, switch sign for those having positive semidefinite Hessian, compute the traces of each Hessian and collect them in the vector $\mathbf{tr}$.
8. Compute the trace heuristic ECO-R as:
$$R^{\text{tr}-\text{heu}} = \boldsymbol{\Psi}\boldsymbol{\omega} \quad, \quad \boldsymbol{\omega} = -\mathbf{tr}/\|\mathbf{tr}\|_2.$$

**Phase 3**

9. *(Optional)* Apply penalization to unexplored state-action pairs.

---
[a]The optimal baseline $b$ is provided in [30, 31].

Alg 1: CR-IRL algorithm.

ing a semidefinite Hessian and switching sign to those with positive semidefinite Hessian. Our heuristic consists in looking for the weights $\boldsymbol{\omega}$ that minimize the trace in this reduced space (in which all ECO-R have a negative semidefinite Hessian). Notice that in this way we can loose the optimal solution since the trace minimizer might assign a non-zero weight to a ECO-R with indefinite Hessian. For brevity, we will indicate with $tr_i = \mathbf{tr}(\mathcal{H}_{\boldsymbol{\theta}} J_i(\boldsymbol{\theta}))$ and $\mathbf{tr}$ the vector whose components are $tr_i$. SDP is no longer needed:

$$\min_{\boldsymbol{\omega}} \quad \boldsymbol{\omega}^T \mathbf{tr} \quad \text{s.t.} \quad \|\boldsymbol{\omega}\|_2^2 = 1. \tag{6}$$

The constraint $\|\boldsymbol{\omega}\|_2^2 = 1$ ensures that, when the ECO-R are orthonormal, the resulting ECO-R has Euclidean norm one. This is a convex programming problem with linear objective function and quadratic constraint, the closed form solution can be found with Lagrange multipliers: $\omega_i = -\frac{tr_i}{\|\mathbf{tr}\|_2}$ (see App. A.2 for the derivation). Refer to Algorithm 1 for a complete overview of CR-IRL (the computational analysis of CR-IRL is reported in App. E).

CR-IRL does not assume to know the state space $\mathcal{S}$ and the action space $\mathcal{A}$, thus the recovered reward is defined only in the state-action pairs visited by the expert along the trajectories in $\mathcal{D}$. When the state and action spaces are known, we can complete the reward function also for unexplored state-action pairs assigning a penalized reward (e.g., a large negative value), otherwise the penalization can be performed online when the recovered reward is used to solve the forward RL problem.

# 6 Related Work

There has been a surge of recent interest in improving IRL in order to make it more appealing for real-world applications. We highlight the lines of works that are more related to this paper.

We start investigating how IRL literature has faced the problem of designing a suitable reward space. Almost all the IRL approaches share the necessity to define *a priori* a set of handcrafted features,

spanning the approximation space of the reward functions. While a good set of basis functions can greatly simplify the IRL problem, a bad choice may significantly harm the performance of any IRL algorithm. The Feature construction for Inverse Reinforcement Learning (FIRL) algorithm [17], as far as we know, is the only approach that explicitly incorporates the feature construction as an inner step. FIRL alternates between optimization and fitting phases. The optimization phase aims to recover a reward function—from the current feature set as a linear projection—such that the associated optimal policy is consistent with the demonstrations. In the fitting phase new features are created (using a regression tree) in order to better explain regions where the old features were too coarse. The method proved to be effective achieving also (features) transfer capabilities. However, FIRL requires the MDP model to solve the forward problem and the complete optimal policy for the fitting step in order to evaluate the consistency with demonstrations.

Recent works have indirectly coped with the feature construction problem by exploiting neural networks [12, 3, 13]. Although effective, the black-box approach does not take into account the MDP structure of the problem. RL has extensively investigated the feature construction for the forward problem both for value function [25, 26, 32, 33] and policy [21] features. In this paper, we have followed this line of work mixing concepts deriving from policy and value fields. We have leveraged on the policy gradient theorem and on the associated concept of compatible functions to derive ECO-Q features. First-order necessary conditions have already been used in literature to derive IRL algorithm [9, 34]. However, in both the cases the authors assume a fixed reward space under which it may not be possible to find a reward for which the expert is optimal. Although there are similarities, this paper exploits first-order optimality to recover the reward basis while the "best" reward function is selected according to a second-order criterion. This allows recovering a more robust solution overcoming uncertainty issues raised by the use of the first-order information only.

# 7  Experimental results

We evaluate CR-IRL against some popular IRL algorithms both in discrete and in continuous domains: the Taxi problem (discrete), the Linear Quadratic Gaussian and the Car on the Hill environments (continuous). We provide here the most significant results, the full data are reported in App. D.

## 7.1  Taxi

The Taxi domain is defined in [35]. We assume the expert plays an $\epsilon$-Boltzmann policy with fixed $\epsilon$:

$$\pi_{\boldsymbol{\theta},\epsilon}(a|s) = (1-\epsilon)\frac{e^{\boldsymbol{\theta}_a^T \boldsymbol{\zeta}_s}}{\sum_{a' \in \mathcal{A}} e^{\boldsymbol{\theta}_{a'}^T \boldsymbol{\zeta}_s}} + \frac{\epsilon}{|\mathcal{A}|},$$

where the policy features $\boldsymbol{\zeta}_s$ are the following state features: current location, passenger location, destination location, whether the passenger has already been pick up.

This test is meant to compare the learning speed of the reward functions recovered by the considered IRL methods when a Boltzmann policy ($\epsilon = 0$) is trained with REINFORCE [22]. To evaluate the robustness to imperfect experts, we introduce a noise ($\epsilon$) in the optimal policy. Figure 2 shows that CR-IRL, with 100 expert's trajectories, outperforms the true reward function in terms of convergence speed regardless the exploration level. Behavioral Cloning (BC), obtained by recovering the maximum likelihood $\epsilon$-Boltzmann policy ($\epsilon = 0$, $0.1$) from expert's trajectories, is very susceptible to noise.

We compare also the second-order criterion of CR-IRL to single out the reward function with Maximum Entropy IRL (ME-IRL) [6] and Linear Programming Apprenticeship Learning (LPAL) [5] using as reward features the set of ECO-R (comparisons with different sets of features is reported in App. D.2). We can see in Figure 2 that ME-IRL does not perform well when $\epsilon = 0$, since the transition model is badly estimated. The convergence speed remains very slow also for $\epsilon = 0.1$, since ME-IRL does not guarantee that the recovered reward is a maximum of $J$. LPAL provides as output an apprenticeship policy (not a reward function) and, like BC, is very sensitive to noise and to the quality of the estimated transition model.

## 7.2  Linear Quadratic Gaussian Regulator

We consider the one-dimensional Linear Quadratic Gaussian regulator [36] with an expert playing a Gaussian policy $\pi_K(\cdot|s) \sim \mathcal{N}(Ks, \sigma^2)$, where $K$ is the parameter and $\sigma^2$ is fixed.

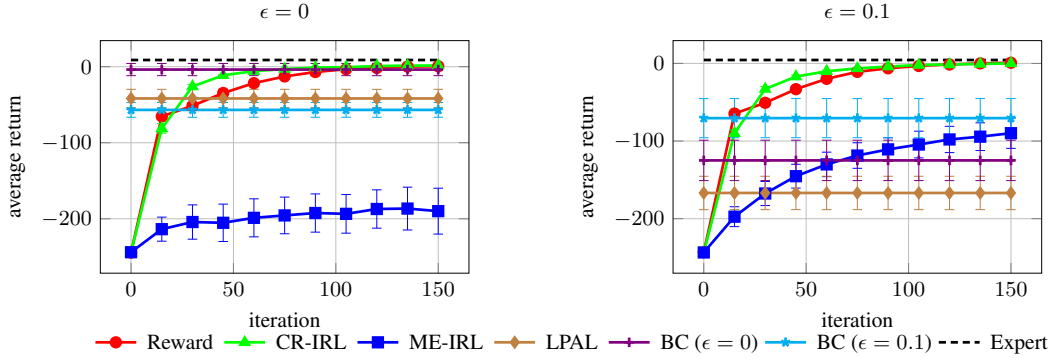

Figure 2: Average return of the Taxi problem as a function of the number of iterations of REINFORCE.

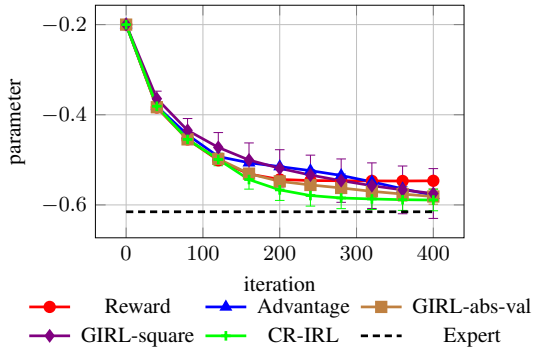

Figure 3: Parameter value of LQG as a function of the number of iterations of REINFORCE.

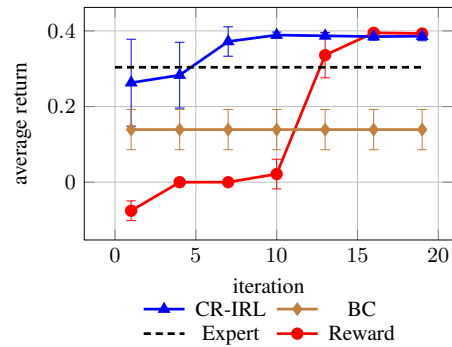

Figure 4: Average return of Car on the Hill as a function of the number of FQI iterations.

We compare CR-IRL with GIRL [9] using two linear parametrizations of the reward function: $R(s, a, \boldsymbol{\omega}) = \omega_1 s^2 + \omega_2 a^2$ (GIRL-square) and $R(s, a, \boldsymbol{\omega}) = \omega_1 |s| + \omega_2 |a|$ (GIRL-abs-val). Figure 3 shows the parameter ($K$) value learned with REINFORCE using a Gaussian policy with variance $\sigma^2 = 0.01$. We notice that CR-IRL, fed with 20 expert's trajectories, converges closer and faster to the expert's parameter w.r.t. to the true reward, advantage function and GIRL with both parametrizations.

### 7.3 Car on the Hill

We further experiment CR-IRL in the continuous Car on the Hill domain [37]. We build the optimal policy via FQI [37] and we consider a noisy expert's policy in which a random action is selected with probability $\epsilon = 0.1$. We exploit 20 expert's trajectories to estimate the parameters $\mathbf{w}$ of a Gaussian policy $\pi_{\mathbf{w}}(a|\mathbf{s}) \sim \mathcal{N}(y_{\mathbf{w}}(\mathbf{s}), \sigma^2)$ where the mean $y_{\mathbf{w}}(\mathbf{s})$ is a radial basis function network (details and comparison with $\epsilon = 0.2$ in appendix D.4). The reward function recovered by CR-IRL does not necessary need to be used only with policy gradient approaches. Here we compare the average return as a function of the number of iterations of FQI, fed with the different recovered rewards. Figure 4 shows that FQI converges faster to optimal policies when coped with the reward recovered by CR-IRL rather than with the original reward. Moreover, it overcomes the performance of the policy recovered via BC.

## 8 Conclusions

We presented an algorithm, CR-IRL, that leverages on the policy gradient to recover, from a set of expert's demonstrations, a reward function that explains the expert's behavior and penalizes deviations. Differently from large part of IRL literature, CR-IRL does not require to specify a priori an approximation space for the reward function. The empirical results show (quite unexpectedly) that the reward function recovered by our algorithm allows learning policies that outperform both behavioral cloning and those obtained with the true reward function (learning speed). Furthermore, the Hessian trace heuristic criterion, when applied to ECO-R, outperforms classic IRL methods.

## Acknowledgments

This research was supported in part by French Ministry of Higher Education and Research, Nord-Pas-de-Calais Regional Council and French National Research Agency (ANR) under project ExTra-Learn (n.ANR-14-CE24-0010-01).

## Footnotes

[1]We want to stress that our primal objective is to recover the reward function since we aim to explain the motivations that guide the expert and to transfer it, not just to replicate the behavior. As explained in the introduction, we aim to exploit the synergy between BC and IRL.

[2]Notice that any linear combination of the ECO-Q also satisfies the first-order optimality condition.

[3]The inner product as defined is clearly symmetric, positive definite and linear, but there could be state-action pairs never visited, i.e., $d_{\mu}^{\pi_{\boldsymbol{\theta}}}(s,a) = 0$, making $\langle f, f \rangle_{\mu, \pi_{\boldsymbol{\theta}}} = 0$ for non-zero $f$. To ensure the properties of the inner product, we assume to compute it only on visited state-action pairs.

[4]A normalization condition is necessary since the magnitude of the trace of a matrix can be arbitrarily changed by multiplying the matrix by a constant.

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
