[Supplementary Material]

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

# A    Proofs and derivations

## A.1    Proof of Proposition 1

*Proof.* $\Gamma_{\pi_\theta} = \{\nabla_\theta \pi_\theta \boldsymbol{\alpha} : \boldsymbol{\alpha} \in \mathbb{R}^k\}$ is a subspace of $\mathbb{R}^{|\mathcal{S}||\mathcal{A}|}$ generated by $k$ vectors, thus its dimension cannot be larger than $k$. For each state $s \in \mathcal{S}$ it holds that $\pi_\theta(\cdot|s)$ is a probability density function thus:

$$\sum_{a \in \mathcal{A}} \pi_\theta(a|s) = 1, \qquad \forall s \in \mathcal{S}.$$

For all parameters $\theta_j$, $j = 1, ..., k$ the partial derivatives of the previous equation result in:

$$\sum_{a \in \mathcal{A}} \frac{\partial \pi_\theta}{\partial \theta_j}(a|s) = 0, \qquad \forall s \in \mathcal{S}, \quad j = 1, 2, ..., k.$$

Let us consider $h \in \Gamma_{\pi_\theta}$, by definition of $\Gamma_{\pi_\theta}$, it can be written as a linear combination of the partial derivatives of $\pi_\theta$:

$$h = \sum_{j=1}^k \alpha_j \frac{\partial \pi_\theta}{\partial \theta_j},$$

so for all the states $s$:

$$\sum_{a \in A} h(a|s) = 0, \qquad \forall s \in \mathcal{S}.$$

These are $|\mathcal{S}|$ linearly independent equations, thus the dimension of $\Gamma_{\pi_\theta}$ cannot be larger then $|\mathcal{S}||\mathcal{A}| - |\mathcal{S}|$. $\qquad \square$

## A.2    Derivation of closed form solution of problem (6)

For two symmetric matrices $A$ and $B$, it holds from Weyl's inequality that $\lambda_{\max}(A) + \lambda_{\max}(B) \leq \lambda_{\max}(A + B)$, thus any linear combination of semidefinite matrices with positive coefficients for the negative semidefinite ones and negative coefficients for the positive semidefinite ones is guaranteed to yield a negative semidefinite matrix. In principle, nothing can be ensured for the indefinite matrices, without looking to the magnitude of the eigenvalues. In the following derivation, we discard ECO-Rs yielding indefinite Hessian and we assume that the remaining yield negative semidefinite Hessian (for the ECO-Rs that yield a positive semidefinite Hessian we just need to switch the sign).

*Proof.* The problem can be solved in closed form using Lagrange multipliers. The Lagragian function is:

$$\mathcal{L}(\boldsymbol{\omega}, \alpha) = \sum_{i=1}^p \omega_i tr_i + \alpha \left( \sum_{i=1}^p \omega_i^2 - 1 \right),$$

where $\alpha$ is the Lagrange multiplier. We impose that partial derivatives of $\mathcal{L}$ w.r.t. $\omega_j$ and $\alpha$ vanish, thus:

$$\frac{\partial \mathcal{L}}{\partial \omega_j} = tr_j + 2\alpha \omega_j = 0, \qquad j = 1, 2, ..., p,$$

$$\frac{\partial \mathcal{L}}{\partial \alpha} = \sum_{i=1}^p \omega_i^2 - 1 = 0.$$

From the first equation, for $\alpha \neq 0$, we obtain an expression of $\omega_j$ as a function of $\alpha$:

$$\omega_j = -\frac{tr_j}{2\alpha}, \qquad j = 1, 2, ..., p.$$

The case $\alpha = 0$ is uninteresting since it is feasible only if all traces are null. By substitution we obtain an expression for $\alpha$:

$$\alpha = \pm \frac{1}{2} \sqrt{\sum_{i=1}^p tr_i^2} = \pm \frac{1}{2} \|\mathbf{tr}\|_2.$$

Since we are looking for a minimum, provided that all traces are non positive, the objective function is minimized for non negative weights:

$$\omega_j = -\frac{tr_j}{\sqrt{\sum_{i=1}^{p} tr_i^2}} = -\frac{tr_j}{\|\mathbf{tr}\|_2}, \qquad j = 1, 2, ..., p.$$

□

# B Model-Based Construction of Reward Features

The relation between the reward and the Q-function is given by the Bellman equation:

$$Q^{\pi_\theta}(s,a) = R(s,a) + \gamma \int_{\mathcal{S}} \int_{\mathcal{A}} \mathcal{P}(s'|s,a)\pi_\theta(a'|s')Q^{\pi_\theta}(s',a')\mathrm{d}s'\mathrm{d}a', \qquad (7)$$

which can be expressed for the finite case in matrix form as $\mathbf{q}^{\pi_\theta} = \mathbf{r} + \gamma \mathbf{P}^{\pi_\theta} \mathbf{q}^{\pi_\theta}$. Similarly to [10], we can obtain the reward function by reversing the equation: $\mathbf{r} = (\mathbf{I} - \gamma \mathbf{P}^{\pi_\theta})\mathbf{q}^{\pi_\theta}$. Thus the set of ECO-R $\{\psi_i\}_{i=1}^{p}$ are obtained by means of the linear mapping $\mathbf{\Psi} = (\mathbf{I} - \gamma \mathbf{P}^{\pi_\theta})\mathbf{\Phi}$ of the ECO-Q $\{\phi_i\}_{i=1}^{p}$. This transformation however requires the knowledge of the transition model $\mathcal{P}$ which in most of the real cases cannot be assumed. As shown in [10], in order to exploit only expert's demonstrations it is necessary to resort to heuristics that may be hard to compute in practice or additional samples for exploration of the system dynamics.

# C Second-Order Optimality Criteria

In this Appendix, we report the SDP formulation of the maximum eigenvalue and trace optimality criteria introduced in Section 5.

## C.1 Maximum eigenvalue optimality criterion

The minimization of the curvature in the flattest direction of the expected return corresponds analytically to the minimization of the maximum eigenvalue of the Hessian. The problem can be formulated using semidefinite programming:

$$\min_{\boldsymbol{\omega}} \quad \lambda_{\max}(\mathcal{H}_{\boldsymbol{\theta}} J(\boldsymbol{\theta}, \boldsymbol{\omega})) \quad \text{s.t.} \quad \mathcal{H}_{\boldsymbol{\theta}} J(\boldsymbol{\theta}, \boldsymbol{\omega}) + \epsilon \mathbf{I} \preceq 0,$$

where $\epsilon \geq 0$ is a threshold to ensure the Hessian is (strictly) negative definite; a further normalization constraint over the weights (like $\|\boldsymbol{\omega}\|_2 = 1$) might be added to ensure boundedness.

This optimization problem is for sure feasible (for sufficiently small $\epsilon$) since, as already seen, the true reward function and the advantage function make the policy parametrization optimal. However, in most of the cases, it is impractical to solve, at least for two reasons. First, the computational effort is enormous even for few policy parameters. Second, it might be the case that the strict negative definiteness constraint is never satisfied due to blocked-to-zero eigenvalues. This problem arises quite often and is related to the presence of "useless" policy parameters that even if modified do not affect the policy performance. In those cases $\lambda_{\max}(\mathcal{H}_{\boldsymbol{\theta}} J(\boldsymbol{\theta}, \boldsymbol{\omega}))$ would be zero no matter which reward weights $\boldsymbol{\omega}$ are selected.

## C.2 Trace optimality criterion

In order to account for the possible presence of blocked-to-zero eigenvalues, we need to consider also the other eigenvalues of the Hessian matrix instead of the maximum one only. Minimizing the trace of the Hessian can be formulated in the following semidefinite programming problem:

$$\min_{\boldsymbol{\omega}} \quad \mathbf{tr}(\mathcal{H}_{\boldsymbol{\theta}} J(\boldsymbol{\theta}, \boldsymbol{\omega})) \quad \text{s.t.} \quad \mathcal{H}_{\boldsymbol{\theta}} J(\boldsymbol{\theta}, \boldsymbol{\omega}) + \epsilon \mathbf{I} \preceq 0,$$

Here the negative definiteness constraint is substantial to ensure the optimality, since there might be ECO-R with negative trace but positive eigenvalues. Moreover, in presence of blocked-to-zero eigenvalues, $\epsilon$ should be set to zero to ensure feasibility.

## C.3 Multi-Objective interpretation of the second-order criteria

The intuitive idea behind the second-order criteria we presented in the previous sections consists in preferring rewards that penalize the most deviations from the expert's policy. This notion can be formalized, in the general case, as the multi-objective SDP problem of minimizing the vector of eigenvalues of the policy Hessian:

$$\min_{\boldsymbol{\omega}} \quad \boldsymbol{\lambda}(\boldsymbol{\omega}) = \big( \lambda_1(\boldsymbol{\omega}), \lambda_2(\boldsymbol{\omega}), ..., \lambda_k(\boldsymbol{\omega}) \big) \quad \text{s.t.} \quad \mathcal{H}_{\boldsymbol{\theta}} J(\boldsymbol{\theta}, \boldsymbol{\omega}) + \epsilon \mathbf{I} \preceq 0,$$

where $\lambda_i(\boldsymbol{\omega}) = \lambda_i(\mathcal{H}_{\boldsymbol{\theta}} J(\boldsymbol{\theta}, \boldsymbol{\omega}))$ for $i = 1, 2, ..., k$ is the $i$-th largest eigenvalue of the policy Hessian. Clearly, among all possible feasible solutions we seek for the (strict) Pareto optimal ones $\boldsymbol{\omega}^P$, i.e., those for which there exists no feasible weight vector $\boldsymbol{\omega}$ such that $\lambda_i(\boldsymbol{\omega}) \leq \lambda_i(\boldsymbol{\omega}^P)$, $i = 1, 2, ..., k$ with at least one strict inequality. It is well-known that the Pareto frontier (the set of all the Pareto optimal solutions) cannot be computed efficiently in most of the cases. A standard approach to tackle this problem is *scalarization* [38]. Scalarization consists in formulating a single-objective optimization problem such that the optimal solutions to the single-objective problem are Pareto optimal solutions to the multi-objective problem and vice versa. A common choice is linear scalarization, that consists in combining the multiple objectives via a linear function:

$$L(\boldsymbol{\lambda}(\boldsymbol{\omega}), \boldsymbol{\beta}) = \sum_{i=1}^{k} \beta_i \lambda_i(\boldsymbol{\omega}) = \boldsymbol{\beta}^T \boldsymbol{\lambda}(\boldsymbol{\omega}). \tag{8}$$

It can be proved that the minimizer of the scalar objective $L(\boldsymbol{\lambda}(\boldsymbol{\omega}), \boldsymbol{\beta})$ is a Pareto optimal solution for every value of $\boldsymbol{\beta} \geq 0$. However, this scalarization is guaranteed to yield all Pareto optimal solutions only when the Pareto frontier is convex, which is not our case as, for instance, $\lambda_k(\boldsymbol{\omega})$ is concave.

It is worth noting that the maximum eigenvalue and the trace optimality criteria fall in this formulation. The former is obtained by setting $\beta_1 = 1$ and $\beta_i = 0$, $i = 2, 3, ..., k$, while the latter is obtained by setting $\beta_i = 1$, $i = 1, 2, ..., k$. Therefore, we are sure that those criteria provide Pareto optimal solutions. Furthermore, the two criteria correspond to popular solution concepts in multi-agent decision theory [39]. The maximum eigenvalue optimality criterion recovers the egalitarian social welfere solution, i.e., the solution that minimizes the maximum unsatisfaction, whereas the trace optimality criterion seeks for the utilitarian solution, i.e., the solution that maximizes the sum of profits. The trace heuristic, however, does not guarantee that the recovered solution is Pareto optimal.

Furthermore, the single-objective function (8) is not convex for all values of $\boldsymbol{\beta}$. It can be proved that a sufficient condition for convexity is that $\beta_1 \geq \beta_2 \geq ... \geq \beta_k \geq 0$ [40], to which both maximum eigenvalue and trace optimality criteria comply.

# D Experimental results

## D.1 Estimators

For the model-based IRL algorithms, the transition model is estimated using the available expert's trajectories via the Monte Carlo estimator:

$$\hat{\mathcal{P}}(s'|s, a) = \frac{\sum_{i=1}^{N} \sum_{t=0}^{T(\tau_i)-1} \mathbb{1}\big( s_{\tau_i, t+1} = s', a_{\tau_i, t} = a, s_{\tau_i, t} = s \big)}{\sum_{i=1}^{N} \sum_{t=0}^{T(\tau_i)-1} \mathbb{1}\big( a_{\tau_i, t} = a, s_{\tau_i, t} = s \big)}. \tag{9}$$

The Kullback-Leibler divergence between the expert's policy $\pi_{\boldsymbol{\theta}}^E$ and the recovered policy $\pi_{\hat{\boldsymbol{\theta}}}$ is estimated from samples as:

$$\hat{d}_{KL}(\pi_{\boldsymbol{\theta}}^E, \pi_{\hat{\boldsymbol{\theta}}}) = \sum_{i=1}^{M} \frac{1}{T(\tau_i)} \sum_{t=0}^{T(\tau_i)} \log \left( \frac{\pi_{\boldsymbol{\theta}}^E(a_{\tau_i, t} | s_{\tau_i, t})}{\pi_{\hat{\boldsymbol{\theta}}}(a_{\tau_i, t} | s_{\tau_i, t})} \right), \tag{10}$$

where the $M$ trajectories are collected with the expert's policy $\pi_{\boldsymbol{\theta}}^E$.

## D.2 Taxi

The environment corresponds to a 5x5 grid in which there are 4 locations, labeled by different letters. The job of the taxi driver is to pick up the passenger at one location and drop him off in another. You

Figure 5: Kullback-Leibler divergence between expert's policy and learned policy in the Taxi problem as a function of the number of iterations of REINFORCE.

receive +20 points for a successful dropoff and lose 1 point for every timestep it takes. There is also a 10 point penalty for illegal pick-up and drop-off actions.

The training of the Boltzmann policy is performed exploiting REINFORCE [22] with Adam [41] (learning rate: 0.008, iterations: 1000).

Besides the average return, shown in section 7.1, we compare the recovered reward functions in terms of distance between the expert's policy and the learned policy (Kullback-Leibler divergence) estimated with (10). Figure 5 confirms the faster convergence of CR-IRL w.r.t. the true reward and ME-IRL. We observe that LPAL is able to recover a policy that is more similar to the expert's policy w.r.t. the one produced by BC, however the average return is worse (this is a consequence of the fact that LPAL does not learn well to perform pick-up and drop-off actions that are associated with the largest rewards).

We also test the model-based version of CR-IRL, where the transition model is estimated with (9). In Figure 6 we notice that the usage of model-based ECO-R instead of model-free ECO-R has no relevant impact on CR-IRL and LPAL, while ME-IRL benefits from the model-based features only when the expert is not noisy.

The trace heuristic can be also used when a given set of features (different from ECO-R) is provided. Since the usage of the Hessian makes sense only when all the considered features are a stationary point of the policy gradient we first need to remove the orthogonal projections over the space spanned by the gradient log-policy. We compare (Figure 7) the learning performance of the set of the first 100 Proto-Value Functions (PVF) [25] when linearly combined with ME-IRL, LPAL and our trace heuristic. We notice that LPAL outperforms both the maximum entropy and the Hessian approaches regardless of the exploration level. Trace heuristic is slightly more effective w.r.t. maximum entropy when the expert is not noisy. Overall the ECO-R yields a better performance w.r.t. the PVFs.

The last set of experiments is aimed to evaluate the effect of the number of expert's trajectories on the average return of the recovered reward functions (Figure 8). The experiment is performed with model-free ECO-R. We notice that CR-IRL is susceptible to the number of expert's trajectories only when $\epsilon = 0.1$: the expert demonstrated a suboptimal behavior and this is more likely when the trajectories are many. This reflects on the estimation of the reward function that does not penalize suboptimal actions performed by the expert. LPAL shows the expected behavior, improving the average return as the number of trajectories increases. Also BC improves overall with the number of trajectories, more effectively when the expert is not noisy.

### D.3 Linear Quadratic Gaussian

The Linear Quadratic Gaussian (LQG) corresponds to the discrete time control problem in a continuous state-action space. The state and reward equations are given by:

$$\mathbf{s}_{t+1} = \mathbf{A}\mathbf{s}_t + \mathbf{B}\mathbf{a}_t + \boldsymbol{\eta}_t \qquad r_t = -\mathbf{s}_t^T \mathbf{Q}\mathbf{s}_t - \mathbf{a}_t^T \mathbf{R}\mathbf{a}_t,$$

Figure 6: Average return of the Taxi problem as a function of the number of iterations of REINFORCE for reward functions recovered from model-based ECO-Rs.

Figure 7: Average return of the Taxi problem as a function of the number of iterations of REINFORCE with the reward function obtained from PVFs.

where $\mathbf{A}$, $\mathbf{B}$, $\mathbf{Q}$ and $\mathbf{R}$ are coefficient matrices and $\boldsymbol{\eta}_t$ is a noise process assumed to be a Gaussian white noise $\boldsymbol{\eta}_t \sim \mathcal{N}(\mathbf{0}, \boldsymbol{\Lambda})$ with uncorrelated components $\boldsymbol{\Lambda} = \lambda^2 \mathbf{I}$. The goal of the agent is to reach as soon as possible the origin since it receives, at each time step, a penalization proportional to the magnitude of the state and the action. The optimal control policy in steady state conditions is the linear controller $\mathbf{a}_t = \mathbf{K}\mathbf{s}_t$ where matrix $\mathbf{K}$ can be found by solving the Riccati equation [36]. The values of the coefficients are reported in Table 1.

Table 1: Coefficients of the LQG experiments. $\mathbf{K}^*$ is the optimal parameter.

| $\mathbf{A}$ | $\mathbf{B}$ | $\mathbf{Q}$ | $\mathbf{R}$ | $\lambda^2$ | $\mathbf{K}^*$ |
|------|------|------|------|------|------|
| 1.0 | 1.0 | 0.9 | 0.9 | 0.1 | $-0.61525$ |

The training is performed exploiting REINFORCE [22] with Adam [41] (learning rate: 0.002, iterations: 600).

We investigate the effect of the number of expert's demonstrations on the performance of the considered IRL algorithms. Figure 9 reports the convergence parameter value and the average return for a variable number of expert's trajectories. We notice that CR-IRL is less sensitive w.r.t. to GIRL since, even if few expert's trajectories are provided, CR-IRL penalizes the regions that the expert did not visit, avoiding the agent to fall into low-reward regions. Differently from the discrete case, in the continuous case the penalization must be "soft", i.e., the more a non-visited state-action pair is far from the visited ones the more it is penalized.

Figure 8: Average return of the Taxi problem as a function of the number of expert's trajectories.

Figure 9: Parameter and average return of LQG as a function of the number of expert's trajectories.

Table 2 and Table 3 show the values of the gradient and the Hessian computed for the different recovered reward functions. GIRL, by construction, yields the reward functions with the smallest gradient in absolute value. On the contrary CR-IRL provides a reward function with larger gradient variance; this is justified by the fact that the Hessian is a large negative number making the policy parameter value a very unstable point for the expected return. Figure 10 shows the average return as a function of the number of iterations for two different values of the policy variance ($\sigma^2$).

Figure 10: Average return of LQG as a function of the number of iterations of REINFORCE.

Table 2: Gradient values of the recovered reward functions of LQG for different values of $\sigma^2$.

| Reward function | $\sigma^2 = 1$ | $\sigma^2 = 0.01$ |
|---|---|---|
| True reward | $-0.09203 \pm 0.5076$ | $0.03185 \pm 0.08893$ |
| Advantage function | $-0.09356 \pm 0.4983$ | $0.9083 \pm 1.467$ |
| CR-IRL | $2.822 \pm 10.77$ | $-0.7514 \pm 55.45$ |
| GIRL-abs-val | $(-2.587 \pm 1.422)\mathrm{e}{-}15$ | $(1.858 \pm 6.068)\mathrm{e}{-}14$ |
| GIRL-square | $(-1.256 \pm 1.534)\mathrm{e}{-}14$ | $(-2.144 \pm 5.232)\mathrm{e}{-}14$ |

Table 3: Hessian values of the recovered reward functions of LQG for different values $\sigma^2$.

| Reward function | $\sigma^2 = 1$ | $\sigma^2 = 0.01$ |
|---|---|---|
| True reward | $-3.064 \pm 7.262$ | $-6.371 \pm 5.847$ |
| Advantage function | $-3.601 \pm 6.521$ | $-223.1 \pm 114.4$ |
| CR-IRL | $-1854 \pm 147.5$ | $-28702 \pm 2586$ |
| GIRL-abs-val | $3.572 \pm 12.33$ | $-17.25 \pm 32.84$ |
| GIRL-square | $-0.3941 \pm 8.218$ | $-6.213 \pm 4.956$ |

We also investigate the effect of the number of neighbors in the extension of the recovered reward function on unvisited state-action pairs. Figure 11 shows the parameter value and the average return as a function of the number of iterations (the bandwidth of the gaussian kernel are fixed to $\sigma_{\mathcal{S}} = \sigma_{\mathcal{A}} = 4$) and different values of the expert's policy variance $\sigma^2$. The KNN interpolation allows reconstructing the reward even in regions of the state-action space which are very far from the region visited by the expert, however, in those regions, the recovered reward is not reliable. We compared the performance of the recovered ECO-R as it is and the performance achieved by penalizing the state-action pairs far from the expert exploration region. In the former case (dashed lines) the convergence is slow and displays a high variance, while in the latter case (solid lines) the learning curve converges faster to the optimal parameter. The number of neighbors has no relevant impact, at least for sufficiently small values, when penalization is applied; while when no penalization is applied the effect is more visible. In particular, when the expert is explorative the recovered reward is smoother (many state-action pairs are visited) so 1NN yields the best performance. As the expert becomes less explorative, a large number of neighbors (50NN) performs better since the recovered reward is more discontinuous. We notice that the penalization becomes more beneficial when the expert's policy is exploitative ($\sigma^2 = 0.01$).

### D.4 Car on the Hill

The policy used to collect samples is a mix of the deterministic optimal policy provided by FQI (ExtraTree estimators: 50, metric: mean squared error) and a random policy:

$$\pi_\epsilon^E(a|\mathbf{s}) = (1 - \epsilon)\pi_{FQI}^E(a|\mathbf{s}) + \epsilon\pi_{\mathrm{random}}(a|\mathbf{s}),$$

where $\epsilon$ is the exploration level. We collected 20 episodes from policy $\pi_\epsilon^E$ and we used them to fit the parameters $\mathbf{w}$ of a Gaussian policy $\pi_{\mathbf{w}}(a|\mathbf{s}) \sim \mathcal{N}(y_{\mathbf{w}}(\mathbf{s}), \sigma^2)$ where $y_{\mathbf{w}}(\mathbf{s})$ is a radial basis function network:

$$y_{\mathbf{w}}(\mathbf{s}) = \sum_{i=1}^{k} w_i e^{-\delta\|\mathbf{s}-\mathbf{s}_i\|^2}. \tag{11}$$

The centers are uniformly distributed in the (position-velocity) state-space (Table 4). The parameters $\mathbf{w}$ are learned by minimizing the KL-divergence between $\pi_{\mathbf{w}}$ the expert's policy $\pi_\epsilon^E$.

We investigate also the effect of the exploration level $\epsilon$ on the performance of the recovered algorithms. Figure 12 shows that CR-IRL outperforms BC for both exploration levels.

In Figure 13 we compare the trajectories of the expert's policy, the maximum likelihood policy (BC) and the policy computed via FQI from the reward recovered by CR-IRL. We can see that when the expert is deterministic the trajectories are almost overlapping. On the contrary, when the exploration rate $\epsilon$ increases we can see that some expert's trajectories fail to reach the profitable absorbing state.

Figure 11: Parameter value and average return of LQG as a function of the number of iterations of REINFORCE with different number of neighbors.

This is a consequence of the fact that a random action is taken with probability $\epsilon$. BC is almost always able to get the final positive reward but not optimally in terms of number of decision epochs required. Finally, CR-IRL, even if trained with a noisy expert, recovers a reward function that induces a policy which is near-optimal, as all trajectories get to the positive final reward in almost the minimum number of steps.

Table 4: Parameters of the Car on the Hill policy.

| $k$ | $\sigma^2$ | $\delta$ |
|---|---|---|
| $40 \times 40$ | 0.01 | 0.01 |

# E   Computational analysis

The main sources of computational complexity in CR-IRL are the SVD, the computation of the Hessian and its eigenvalue computation for each ECO-R. For a $m \times n$ matrix it is well known that SVD has complexity $O(\min\{mn^2, nm^2\})$. We apply SVD twice: the first time it is applied to matrix $\nabla_{\boldsymbol{\theta}} \log \pi_{\boldsymbol{\theta}}{}^T \mathbf{D}_{\mu}^{\pi_{\boldsymbol{\theta}}}$ of dimension $k \times |\mathcal{D}|$ and the second time to matrix $\boldsymbol{\Psi}$ of dimension at most $|\mathcal{D}| \times |\mathcal{D}|$. Assuming $k \leq |\mathcal{D}|$, the second application dominates yielding a complexity of $O(|\mathcal{D}|^3)$. The computation of the Hessian is linear in the number of ECO-R (at maximum $|\mathcal{D}|$) and in the number of samples ($|\mathcal{D}|$) and quadratic in the number of parameters $k$, thus the complexity is $O(k^2|\mathcal{D}|^2)$. Finally the computation of the eigenvalues is cubic in the order of the matrix ($k$) and has to be done for every ECO-R, yielding a cost of $O(k^3|\mathcal{D}|)$, which is dominated by $O(k^2|\mathcal{D}|^2)$ as $k \leq |\mathcal{D}|$. Overall the complexity of CR-IRL is $O(\max\{k^2|\mathcal{D}|^2, |\mathcal{D}|^3\})$.

Figure 12: Average return of Car on the Hill problem varying the value of exploration $\epsilon$.

Figure 13: Trajectories of the expert's policy, the ML policy and the policy computed via FQI from the reward recovered by CR-IRL for different values of $\epsilon$.