[Reviews · NeurIPS 2017]

Reviewer 1



This paper extends compatible value function theorem in RL for constructing reward function features in IRL. Specifically, the main contributions of this paper are: (1) ECO-Q, which constructs the feature space of the (unknown) Q-function by computing the null space of the policy gradient, (2) ECO-R, which constructs the feature space of the reward function by applying Bellman equation to the feature space of the Q-function. Overall, the proposed method is sound and has several advantages. It does not require environment dynamics and solving the forward problem. Going further from the existing work considering first-order necessary condition, it also exploits second order criterion which penalizes the deviation from the expert's policy. Finally, the algorithm generates feature functions of reward (or value) automatically, which eliminates the need to hand-crafted feature. In summary, this paper tackles the continuous control IRL in a systematic way and provides promising results in experiments. I vote for acceptance of this paper.

Reviewer 2



This paper proposes an approach for behavioral cloning that constructs a function space for a particular parametric policy model based on the null space of the policy gradient. I think a running example (e.g., for discrete MDP) would help explain the approach. I found myself flipping back and forth from the Algorithm (page 6) to the description of each step. I have some lingering confusion about using Eq. (4) to estimate discounted vectors d(s,a) since (4) only computed d(s). I assume a similar estimator is employed for d(s,a). My main concern with the paper’s approach is that of overfitting and lack of transfer. The main motivation of IRL is that the estimated reward function can be employed in other MDPs (without rewards) that have similar feature representations. Seeking to construct a sufficiently rich feature representation that can drive the policy gradient to zero suggests that there is overfitting to the training data at the expense of generalization abilities. I am skeptical then that reasonable generalization can be expected to modifications of the original MDP (e.g., if the dynamics changes or set of available actions changes) or to other MDPs. What am I missing that allows this approach to get beyond behavioral cloning? My understanding of the Taxi experiment in particular is that data is generated according to the policy model under line 295 and that the proposed method uses this policy model, while comparison methods are based on different policy models (e.g., BC instead estimates a Boltzmann policy). This seems like an extremely unfair advantage to the proposed approach that could account for all of the advantages demonstrated over BC. What happens when likelihood is maximized for BC under this actual model? Experiments with data generated out of model and generalization to making predictions beyond the source MDP are needed to improve the paper. In summary, though the method proposed is interesting, additional experiments that mimic “real-world” settings are needed before publication to investigate the benefits of the approach suitability to noisy data and for generalization purposes.

Reviewer 3



This paper developed a model-free inverse reinforcement learning (IRL) algorithm to recover the reward function that explain the expert demonstrations. Different from earlier work, the method developed in the paper (CR-IRL) automatically constructs the basis functions for the reward function by exploiting the orthogonality property between the policy and value spaces. The method also uses a second-order condition to rank the reward functions. The method is novel and the presentation of the paper is clear. Here are a few questions to be clarified by the authors: *The method developed in the paper is a batch method, i.e., it applies method like SVD to the statistics over the entire dataset. This approach may have some scalability issue when considering problems with large amount of data. Is it possible to extend the method so that it is more scalable by using algorithms like stochastic gradient descent? Or it is possible to extend to the streaming data case? *It is interesting to observe that the CR-IRL outperforms the true reward function in terms of convergence speed. Could the authors discuss more about the reasons?